# The Impact of COVID-19 Crisis on the Control and Management of Radiography Practice in the United Arab Emirates

**DOI:** 10.3390/healthcare10081546

**Published:** 2022-08-15

**Authors:** Suliman Salih, Ajnas Alkatheeri, Bashayer Almarri, Nouf Al Shamsi, Osama Jaafari, Majedh Alshammari

**Affiliations:** 1Department of Radiography and Medical Imaging, Fatima College of Health Sciences, Al Ain 33003, United Arab Emirates; 2National Cancer Institute, University of Gezira, Wad Madani 2667, Sudan; 3Royal Commission Medical Center, Yanbu 46451, Saudi Arabia; 4Radiology Department, Najran University, Najran 61441, Saudi Arabia

**Keywords:** influence of COVID-19, radiographers, working environment, PPE

## Abstract

The present study aimed to assess the impact of the COVID-19 crisis on radiology practices in Abu Dhabi, UAE. An electronic survey (Google form) was distributed among Abu Dhabi government and private hospitals. The survey included general X–ray services, which were only provided in the radiology departments. The diagnostic radiographers who reported changes in the number and type of radiology procedures (37%) reported that the changes reached 61–80% compared to the number of procedures being conducted prior to the outbreak of COVID-19. While infection control was challenging due to the shortage of personal protective equipment (PPE), 51.2% of the participants were affected. The healthcare workers in the radiology departments in Abu Dhabi are exposed to a high number of COVID-19–infection patients, which increases their chances of contracting the disease. A total of 90% of employees were infected with COVID-19 during the crisis. COVID-19 has resulted in changes in clinical working patterns, such as the type and number of procedures performed daily. Additionally, PPE shortages, staff infection during the pandemic, an increase in workplace–related difficulties, and staff well–being are common consequences of the pandemic. It is vital to enhance coping strategies in order to support staff well–being. However, the psychological effects caused as a result of the pandemic should not be ignored, and providing professional support to workers is recommended.

## 1. Introduction

The coronavirus is known as severe acute respiratory syndrome–related coronavirus species (SARS-CoV-2); it belongs to the genus beta and subgenus sarbecovirus, and it can infect humans and animals. Generally, SARS-CoV-2 consists of an enveloped, large, positive, single RNA genome and, at the morphology level, it has spherical virions with core–shell and surface projections resembling a solar corona with average diameters of 64.8 ± 11.8, 85.9 ± 9.4, and 96.6 ± 11.8 nm (average ± SD) for the short, medium, and long axes of the envelope, respectively. There are four SARS-CoV-2 subfamilies: alpha, which causes mild symptoms; beta, which can cause mortality; gamma; and delta coronaviruses. SARS-CoV-2 presents symptoms in patients that include a fever, dry cough, fatigue, dyspnea, sore throat, sneezing, and diarrhea (focused role) in some cases. As the disease rapidly spread around the globe, it caused public health officials and governments to apply various measures to control it, including travel bans, the imposition of large–scale curfews, isolation, the quarantining of infected individuals, the imposition of remote–work systems, the wearing of face masks in public places, strict social distancing protocols, and restrictions on large gatherings of individuals. As a result, it has had an impact on many aspects of life, industries, imports, exports, and trading activities as well as our global community’s socioeconomic status [1,2,3,4].

It has been well known that the healthcare sector was very severely affected during the pandemic; it was challenging for first–line healthcare providers to continue to provide healthcare without adequate access to PPE. For instance, Italian healthcare providers faced a lack of PPE, leading to high infection and death rates. While the number of confirmed cases has increased globally, the amount of PPE available is not enough to protect the workers. In the US, N95 respirators were used; they were not approved by the Food and Drug Administration (FDA) but received approval from the National Institute for Occupational Safety and Health. Controversially, the Center for Disease Control and Prevention (CDC) recommended the reuse of face masks and respirators intended for one–time use as well as the use of scarves or bandanas if stocks were fully depleted [5,6,7].

The radiology department plays a crucial role in the diagnosis and follow–up of COVID-19 cases. This central role resulted in an increase in work effort and challenges for radiology staff during the pandemic. Radiology has been used to clinically diagnose COVID-19 cases due to its high sensitivity and greater ability to obtain low false–negative rates compared to RT–PCR. Numerous studies have been conducted to measure the application of radiography worldwide during the COVID-19 pandemic by conducting a survey. For example, in Australia, 47.6% of radiographers had access to PPE, which was adequately available, while 20.5% of Middle Eastern, North African, and Indian (MENAIN) radiology staff did not have access to it. Diagnostic radiography (X–ray) workload increased during the pandemic in Australia and MENAIN by 25.6% and 48.1%, respectively. Additionally, the breakout of COVID-19 had a psychological effect on healthcare providers in Australia. The personal stress, anxiety, and stress experienced by the workers’ families, partners, and friends was 61.4%, 58.2%, and 57.4%, respectively. In MENAIN, 42.9% of the participants began to experience work–related stress, 57.1% needed professional help to cope with their stress, and 72.9% of their families were affected by their work–related stress [8,9]. Numerous other studies conducted in Europe and Africa found that the pandemic posed significant challenges to radiographers’ professional practice in a variety of ways [1,5,6,10,11,12,13,14]. A study conducted in Jordan presented the results distribution obtained from a computed tomography (CT) scanner and magnetic resonance imaging (MRI) units in some countries /per million population, concluding that in some countries, such as Jordan, the amount of diagnostic imaging equipment does not correlate with the population number, which leads to an overload of work in many of radiology units [15]. 

To the best of the author’s knowledge, this study is the first to survey the impact of the COVID-19 pandemic on the work environment and well–being of radiographers in Abu Dhabi, UAE. This study discusses, in general terms, the impact of the waves of the COVID-19 pandemic on the mental health and well–being of diagnostic radiographers in the Emirate of Abu Dhabi. This study aims to assess the impact of the COVID-19 crisis on the care of radiology patients and the operation of radiology practices in both public and private clinics in the UAE. These effects appear to be different due to the geographical location, structural conditions, and socioeconomic status of the countries.

## 2. Materials and Methods

### 2.1. Study Design

A cross–sectional survey of radiographers practicing in the Emirate of Abu Dhabi was conducted to assess the workload, PPE availability, staff infected by COVID-19, transferred staff, short–time work, online tools, and essential materials stock. The study was conducted during the period of 17 January to 6 April 2022.

An electronic survey (Google form) was distributed among the Abu Dhabi government and private hospitals. The survey targeted the radiology departments, focusing on the staff performing X–rays. The survey contained an introduction to explain our aim for conducting the survey. The survey was distributed by a link to the official radiology department manager’s email, LinkedIn, and personal contacts. The online survey contained some general practice questions, such as the practice location, expertise, role in the department, and other questions related to various topics to measure the impact of COVID-19 on the radiology departments in Abu Dhabi. This instrument was developed by the authors of the present study.

The participants included healthcare providers working in radiology departments (X–ray) during the pandemic, such as radiological technology supervisors, radiological technology specialists, radiological technicians, and intern radiographers. We used the Cochran formula to calculate the sample size with a 95% confidence and 5% margin of error as well as the study population size of 49 (hospitals); we concluded that 43 (hospitals) participants were required, and we received 46 responses from government and private hospitals and a radiology clinic in Abu Dhabi. The study used simple random sampling, so each radiographer was fairly selected from the radiology departments, thus producing an unbiased sample.

Ethical approval for the study was obtained from the Fatima College of Health Sciences Research Ethics Committee. Additionally, informed consent was obtained once the participants agreed to complete the questionnaire.

### 2.2. Data Analysis

IBM SPSS version 28.0.0.0 (190) and Excel version 22.0, as well as descriptive statistics, were used to analyze the data. Percentages were utilized to describe the overall number of practitioner responses to key variables. A *p*–value of less than 0.05 was the level of statistical significance used throughout the analysis.

## 3. Results

### 3.1. Radiology Procedure Changes 

The survey recorded 43 valid responses from 48 hospitals in Abu Dhabi to measure the radiology department’s response rate during the pandemic. According to Table 1, the majority of respondents (81.4%) are diagnostic radiographers with HAAD licenses, radiologic technology supervisors (7.0%), and department heads (4.7%). As presented in Table 2, the majority of the participants’ expertise is in radiography, working as technicians and radiology technology specialists ((51.2%) and (25.6%), respectively). The government sector accounted for 74.4% of the survey’s participants, while the private sector accounted for 25.6% (Table 3).

The type and number of radiographic procedures significantly changed during the pandemic; 67.7% of diagnostic radiographers agreed that the number of radiographic procedures changed due to COVID-19 (Figure 1). A total of 30.23% assumed that it had changed 41–60%, while 37.21% assumed that the change was around 61–80% (Figure 2). There is a significant positive relationship between COVID-19 and an increase in the number of outpatients in radiology departments (r (39) = 0.56, *p* = 0.001), which provides evidence of an increase in the workload in radiology departments during the COVID-19 crisis. A total of 67.4% of the respondents reported that there was a shift in the relationship between outpatients and stationary patients during the COVID-19 crisis (Figure 3). A total of 72.9% assumed that the proportion increased from 50% to 100%, while 5.4% assumed that there was no change in the proportion (Figure 4).

### 3.2. Impact of COVID-19 on Professional Practice

The COVID-19 crisis had an impact on diagnostic radiographers’ professional practice. The impact included changes in the working hours, transfers to other clinics and areas, and an overload of disease cases due to infected employees in the departments. The majority of participants (76.7%) responded that they did not work for a short period of time during the COVID-19 pandemic, while 7% had their working hours reduced by more than 70% (Figure 5), and around 55.8% had employees from their department transferred to other clinics or areas (Figure 6). Approximately 90% of departments included colleagues infected with COVID-19 during the crisis compared to just 9.3% who did not contain infected staff (Figure 7).

### 3.3. Infection Control: PPE Shortage, PPE in Stock, and Essential Materials Order

All Abu Dhabi hospitals surveyed in the present study followed infection prevention and control (IPC) to prevent the spread of SARS-CoV-2 among the patients and their contacts by minimizing person–to–person transmission in healthcare settings, including all medical staff who frequently interacted with patients, whether infection was suspected or real. The use of disposable work hats, anti–fog face shields or goggles, medical protective masks, isolation gowns or protective clothing, disposable latex gloves, and disposable shoe covers, as well as thoroughly performed hand hygiene techniques, was employed. All the radiology equipment was cleaned using wipes containing 75% ethanol for each patient in each radiology department, which included at least one supervisor to oversee the radiography department’s cleaning and safety protocols. Following patient examinations for confirmed instances and in advance of patient examinations for subsequent suspected cases, the contaminated, semi–polluted, and cleaning areas were strictly segregated and terminally disinfected. The healthcare workers in the radiology department were asked about access to personal protective equipment during the COVID-19 pandemic; around half of the workers (51.2%) had a shortage of access to PPE in their workplace, and the other half (48.8%) had an adequate availability of PPE. In Table 4, a significant negative relationship between increasing the number of radiology procedures due to the COVID-19 crisis and having a shortage of PPE in radiology departments is presented (r (41) = −0.31, *p* = 0.005; Table 5). At present, the stock of PPE is sufficient for 14–28 days as 27.9% and (9.3% for less than 2 days.) Forty–one percent of respondents’ departments are still ordering 20–70% of their normal essential materials. A total of 29.3% are still ordering more than 70% of their normal essential materials. A total of 17.1% of the departments are not making any changes to the order of essential materials. There is a significant negative relationship between changing the number of radiology procedures being conducted due to the COVID-19 crisis and changes in the ordering of radiology essential materials (r (41) = −0.43, *p* = 0.005; Table 6).

### 3.4. COVID-19 and Online Tools

Some of the impacts of COVID-19 on the daily duties being performed in the radiology departments were addressed by using technology that had not been previously used. More than half (58.2%) of the individuals used the online tools for different purposes, while 41.9% did not. The use of online conferences was observable in 27.9% of radiology departments, 23.3% moved to online reporting methods, and 7% conducted radiology consultations online (Table 7).

## 4. Discussion

This study explored the perspectives of radiographers regarding the impact of COVID-19 on clinical radiography practices in Abu Dhabi, UAE. In accordance with the international guidelines [16,17], imaging, specifically CXR and CT, remained the primary diagnostic and management tools used for COVID-19 in Abu Dhabi. To the best of the author’s knowledge, this study was the first to survey UAE–Abu Dhabi city healthcare workers in the radiology department regarding the impact of COVID-19 on their practice. The demographic data showed that the radiology departments in the government sector (74.4%) had higher responses than the private sector (25.6%) (Table 3). The percentage of this study’s respondents who identified as radiographers was 81.4%, supervisors made up 7.0%, and heads of departments made up 4.7% (Table 1).

The results of this study show that COVID-19 had a remarkable impact on the diagnostic radiographers’ professional practice. The impact included changes in work hours, transfers to other clinics and areas, and patient overload due to infected employees in the departments. The majority of participants (76.7%) responded that they did not work for a short period of time during the COVID-19 pandemic, while 7.7% had their working hours reduced by more than 70% (Figure 5); around 55.8% had employees from their department transferred to other clinics or areas (Figure 6). Approximately 90% of departments contained colleagues infected with COVID-19 during the crisis compared to just 9.3% who did not have infected staff in their departments (Figure 7). This result agrees with many studies that present the changes in the work environment and X–ray procedures that occurred during the pandemic [7,10,12,18,19,20,21,22]. These studies confirm the effects of the pandemic on different levels, according to the magnitude of the pandemic and the availability of local resources [6,14,20,21,22,23]. Additionally, our results indicate that the patterns of work change during the pandemic. Some of the impacts of COVID-19 on the daily duties of individuals working in radiology departments was addressed by using technology that has never been used before. More than half (58.2%) of the surveyed departments used online tools for different purposes, while 41.9% did not. Online conferences were used in 27.9% of the radiology departments; 23.3% moved to online reporting; and 7% conducted radiology consultations online (Table 7). This result is in line with the other studies that investigated the changes in daily work patterns in both clinical and management aspects [11,24].

It has been shown that there was a shortage in PPE stock and the ordering of critically needed materials. Around half of the surveyed departments, 51.2%, had a shortage of access to PPE in their workplace. The other half, 48.8%, had adequate availability of PPE. There was a significant negative relationship between increasing the number of radiology procedures due to COVID-19 and having a shortage of PPE in the radiology departments (r (41) = 0.31, *p* = 0.05). At present, 27.9% of the stock of PPE is sufficient for 14–28 days, and 9.3% is sufficient for less than two days. A total of 41% of respondents’ departments are still ordering 20–70% of their normal essential materials order activity. A total of 29.3% of departments are still ordering more than 70% of their normal order activity for essential material. A total of 17.1% of departments did not making any changes to the ordering of essential materials (Table 4). There is a significant negative relationship between changing the number of radiology procedures due to the COVID-19 crisis and changing the ordering of essential materials in the radiology department; r (41) = −0.43, *p* = 0.005. This result is consistent with many other studies that observed a shortage of PPE and workplace–related stress among radiographers due to a fear of contracting the virus, the perceived inadequacy of PPE, and authorities’ relatively weak response to concerns about staff testing. These findings are in agreement with several studies that reported on the infection control issues and consequences related to the pandemic [5,7,17,25,26,27].

## 5. Conclusions

This survey highlighted the impact of COVID-19 on radiography practices in terms of radiology procedure changes, impact on professional practice, infection control, PPE shortage, ordering of essential materials, and utilization of online tools. COVID-19 in public and private hospitals in the Emirate of Abu Dhabi during the pandemic was observed in this study. The pandemic resulted in changes to clinical working patterns, such as types of patients and the number of procedures performed daily that changed from 41 to 60%. Therefore, 51.2% of the participants assumed they faced PPE shortages and 76.7% assumed increased working hours, which added more workplace–related difficulties. Additionally, the impact of the pandemic on the well–being of radiographers included contracting infections as well as stress related to work. It is therefore critical for radiology departments to recognize the need to protect all their staff, including the radiography workforce, to ensure patient safety by providing adequate training, appropriate PPE, and strengthening institutional structures for the management of workplace–related stress and anxiety in similar, future pandemics.

## 6. Limitations of the Study

One limitation of this study was a possible sampling bias. The simple random technique can introduce bias when the sample set is not large enough to adequately represent the entire population. Due to the uncooperative nature of many healthcare institutions that did not participate in the survey, each healthcare worker in the radiology department was used to represent the area of practice.

## Figures and Tables

**Figure 1 healthcare-10-01546-f001:**
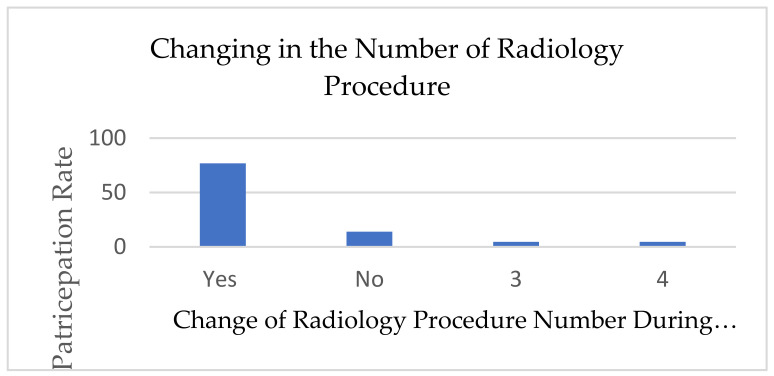
Change in radiology procedure within diagnostic radiography practice due to COVID-19.

**Figure 2 healthcare-10-01546-f002:**
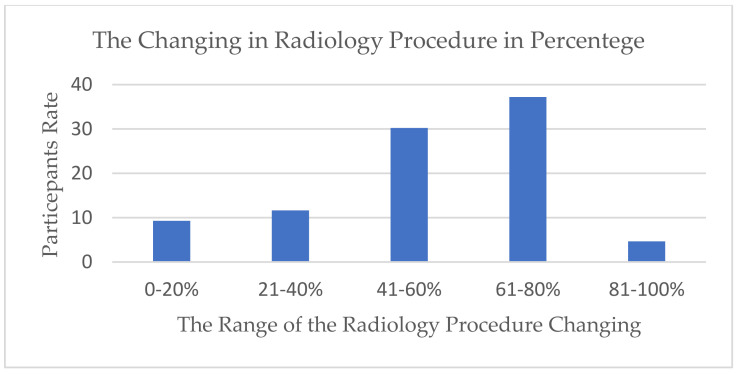
The percentage of change in radiography procedures due to COVID-19.

**Figure 3 healthcare-10-01546-f003:**
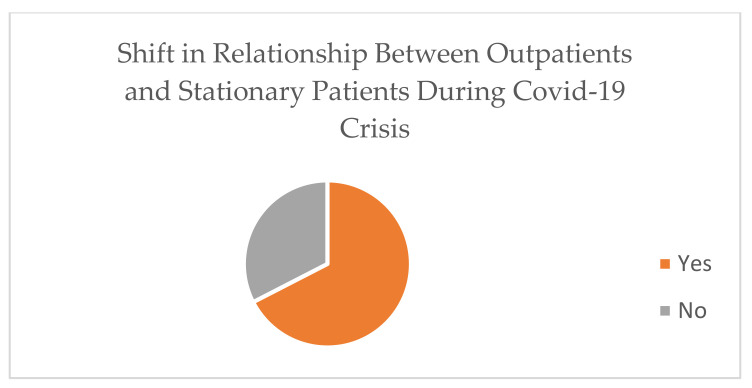
Shift in relationship between outpatients and stationary patients during COVID-19 crisis.

**Figure 4 healthcare-10-01546-f004:**
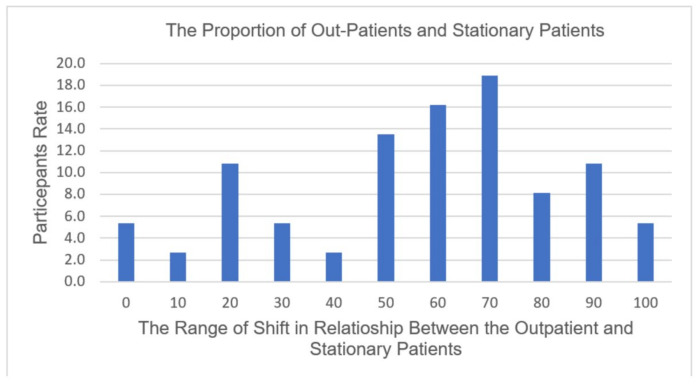
A total of 72.9% of participants assumed that the proportion increased from 50% to 100%.

**Figure 5 healthcare-10-01546-f005:**
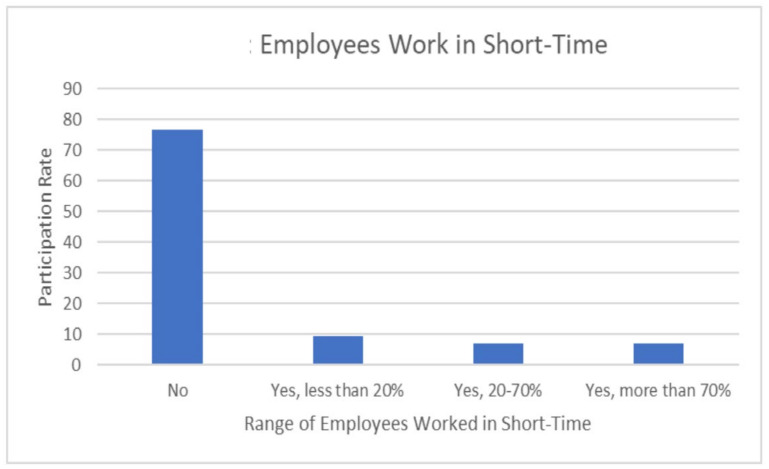
Radiology workers and short–time duty during the COVID-19 crisis.

**Figure 6 healthcare-10-01546-f006:**
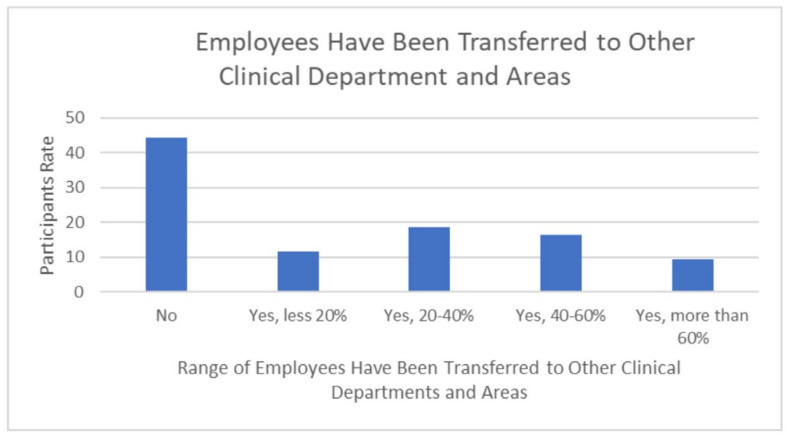
Radiology workers transferred to other clinics or areas.

**Figure 7 healthcare-10-01546-f007:**
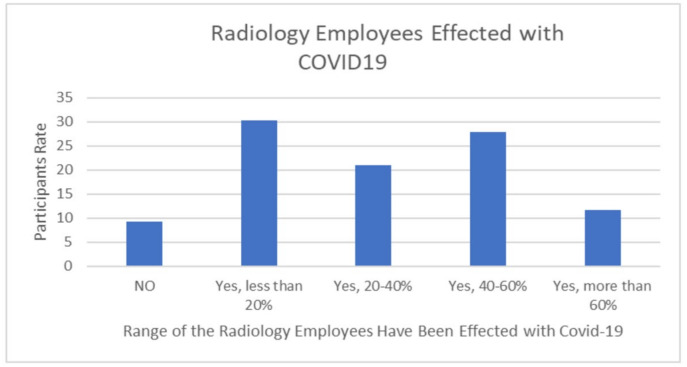
Radiology workers who infected with COVID-19.

**Table 1 healthcare-10-01546-t001:** Function or role in the department.

Primary Profession	Frequency	Percentage %
Head of department	2	4.7%
Consultant	1	2.3%
Resident	1	2.3%
Radiologic technology supervisor	3	7.0%
Radiographer	35	81.4%
Other	1	2.3%

**Table 2 healthcare-10-01546-t002:** Expertise.

Expertise	Frequency	Percentage %
Radiologist	1	2.3%
Radiology technology specialist	11	25.6%
Radiology technician	22	51.2%
Intern radiographer	7	16.3%
Other	2	4.7%

**Table 3 healthcare-10-01546-t003:** Practice sector.

Practice Sector	Frequency	Percentage %
Government hospital	32	74.4%
Private hospital	11	25.6%

**Table 4 healthcare-10-01546-t004:** PPE shortage, stock, and ordering of essential materials in radiology departments.

Infection Control:		Participation Rate%	
PPE	Was there a shortage in PPE during the COVID-19 crisis?	Yes	51.2
		No	48.8
	For how many days do you have PPE in stock for you and your employees?	Less than 2 days	9.3
		2–7 days	23.3
		7–14 days	20.9
		14–28 days	27.9
		More than 28 days	18.6
Essential materials	Did you adjust your orders for essential materials?	No	17.1
		Yes, still ordering >70% of normal activity	29.3
		Yes, still ordering 20–70% of normal activity	41.5
		Yes, still ordering less than 20% of normal activity	12.2

**Table 5 healthcare-10-01546-t005:** Correlation between the number of radiology procedures being conducted due to COVID-19 crisis and PPE shortage in radiology departments.

		Number of Radiology Procedures during COVID-19 Crisis	PPE Shortage in Radiology Departments during COVID-19 Crisis
Number of radiology procedures during COVID-19 crisis	Pearson Correlation	1	0.131
	Sig. (2–tailed)		0.404
	N	43	43
PPE shortage in radiology departments during COVID-19 crisis	Pearson Correlation	0.131	1
	Sig. (2–tailed)	0.404	
	N	43	43

**Table 6 healthcare-10-01546-t006:** Correlation between the number of radiology procedures being conducted due to COVID-19 crisis and changes in ordering of essential materials in radiology departments.

		Number of Radiology Procedures during COVID-19	Ordering Essential Materials during COVID-19
Number of radiology procedures being conducted during COVID-19	Pearson Correlation	1	−0.431 **
	Sig. (2–tailed)		0.005
	N	43	41
Ordering essential materials during COVID-19	Pearson Correlation	−0.431 **	1
	Sig. (2–tailed)	0.005	
	N	41	41

** Correlation is significant at the 0.01 level (2–tailed).

**Table 7 healthcare-10-01546-t007:** Using online tools in radiology department during COVID-19 crisis.

	Participants %
No	41.9
Yes, online conference	27.9
Yes, online reporting	23.3
Yes, video consultations for patients and referring physicians	7.0

## Data Availability

Not applicable.

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
