# Peer review of "The Impact of COVID-19 Crisis on the Control and Management of Radiography Practice in the United Arab Emirates"

_healthcare, 2022, doi:10.3390/healthcare10081546_

Round 1
Reviewer 1 Report
Comments for the Authors:
1. Title is too short and require modification to be comprehensive.
2. Moreover, UAE in title should be define since it mentioned for the first time.
3. Please note that the conclusion in the abstract section is absence. So, kindly rewrite the conclusion line in the abstract according to the manuscript findings.
4. Further references should be added to the Introduction part. You can use the following published articles:
- Focused role of nanoparticles against COVID-19: Diagnosis and treatment
- Assessment of Diagnostic Imaging Sector in Public Hospitals in Northern Jordan
5. The caption of figure 4 should be rewrite.
6. Table 6 is confused. Could you please insert borders between numbers?
7. Please write the IRB number (ethical approval number) in method part. In other words, Institutional Review Board should have a number.
8. Please add more explanations in the discussion section to be fit with the results.
9. Conclusion section is very short. Please explain further results.
10. Please send your manuscript to native speakers to make proofreading because there is several grammatical and typo errors in whole article.
11. Check the values in whole manuscript.

Reviewer 2 Report
The authors present an interesting manuscript with the results of a well-conducted experimentation.
A limitation of this study is that the sample set was not large enough to adequately represent the entire population.
The authors submitted a manuscript to assess the COVID-19 crisis impact on the radiology practices in Abu Dhabi, UAE. The manuscript refers to a small group of radiologists and includes only the X ray service in the radiology departments. In reality, the SARS CoV2 pandemic has mainly involved the CT service in radiology, with all the problems related to patient distancing, sanitation of environments, etc. The problem of infections among radiologists should be better investigated by specifying the type of environmental protection protocols and health workers adopted during radiological examinations of COVID 19 patients. Therefore, the work, even if limited to a small sample, is interesting but some aspects need to be further specified.Author Response
Please see the attachment

Round 2
Reviewer 2 Report
nothing